# Changes in planned and unplanned canopy openings are linked in Europe's forests

Rupert Seidl [1,2] ✉ & Cornelius Senf[1,3]

Canopy openings are increasing in Europe's forests, yet the contributions of anthropogenic and ecological agents of disturbance to this increase remain debated. Here we attribute the root cause of all stand-replacing canopy disturbances identified for Europe in the period 1986–2020 from Landsat data (417,000 km²), distinguishing between planned and unplanned canopy openings (i.e., disturbance by human land use versus by wind, bark beetles, and wildfire). We show that canopy openings by humans dominate the European forest disturbance regime, accounting for 82% of the area disturbed. Both planned and unplanned canopy openings increased in the early 21st century (+24% and +30% relative to the late 20th century). Their changes are linked, with simultaneous increases in planned and unplanned canopy openings on 68% of Europe's forest area. We conclude that an important direction for tackling disturbance change in policy and management is to break the link between planned and unplanned canopy openings in Europe's forests.

Disturbance regimes are changing around the world, altering the forests of the Earth. Disturbances, here defined as pulses of canopy openings[1], can result from either ecological or anthropogenic causes. Ecological disturbances, i.e., canopy openings from agents such as wildfire, windthrow, and insect outbreaks, are among the most climate-sensitive processes in forest ecosystems, and are responding strongly to ongoing changes in the climate system[2,3]. In addition, human land-use is intensifying in many parts of the globe, to meet the demands for renewable resources from a growing human population[4]. Distinguishing these two broad categories of disturbance is important for addressing disturbance change in management, as one results from targeted human activities (i.e., planned canopy openings such as tree harvesting for resource extraction), while the other is highly stochastic and driven strongly by climatic extremes (i.e., unplanned canopy openings from ecological agents such as wildfire or insect activity). Changes in both planned and unplanned canopy openings can profoundly alter forest ecosystems, leading to forests that are more open as well as comprised of younger and smaller trees[5]. Such changes in the structure and functioning of forest ecosystems can have distinct implications for the benefits humans derive from forests. Increasing disturbances could, for instance, reduce the ability of forests to store carbon, impairing the contribution of forests to climate change mitigation[6,7]. Addressing disturbance change is thus one of the biggest challenges for current forest policy and management.

In Europe, forest disturbance change is a topic of intensive debate. This debate centers on the question whether anthropogenic or ecological drivers are primarily responsible for the observed increase in canopy openings[8–10]. Answering this question and determining the degree to which different disturbance types are linked is important for addressing disturbance change in forest policy and management. Here, we present evidence that both planned and unplanned canopy openings are increasing in Europe's forests and that their increase is linked. We attribute each individual high severity disturbance patch identified in a satellite-based disturbance map for Europe (ground resolution: 30 m[11]), for the period 1986 to 2020 to either planned (i.e., land-use) or unplanned (i.e., fire, wind, and bark beetles) causes. Attribution is based on an existing algorithm[12,13], which we update to include bark beetle disturbances, and extend to cover the most recent period of intensive disturbance activity in Europe (see "Methods" for details). We analyze patterns in planned and unplanned canopy openings at the level of administrative units nested within countries (*Nomenclature des unités territoriales statistiques* level 2 [NUTS2]), with

[1]Technical University of Munich, TUM School of Life Sciences, Ecosystem Dynamics and Forest Management, Freising, Germany. [2]Berchtesgaden National Park, Berchtesgaden, Germany. [3]Technical University of Munich, TUM School of Life Sciences, Earth Observation for Ecosystem Management, Freising, Germany. ✉e-mail: rupert.seidl@tum.de

a mean size of 17,647 km² (sd ± 22,927 km²). We focus our work on all countries predominantly situated in continental Europe but exclude Belarus, Bosnia and Herzegovina, Kosovo, Moldova, and Ukraine, for which no NUTS2 classification was available. We henceforth refer to our study area as Europe for the sake of readability.

## Results

### Planned and unplanned canopy openings in Europe's forests

Canopy openings from human land use strongly dominate the disturbance regime of Europe's forests. A total canopy area of 417,000 km² was disturbed between 1986 and 2020, whereof 82.2% was attributed to planned canopy openings related to human land use. Unplanned canopy openings (i.e., by wind, bark beetles and wildfire) were jointly responsible for 2118 km² of disturbance per year on average between 1986 and 2020. The dominance of anthropogenic disturbance was particularly strong in northern and eastern Europe, while ecological disturbances dominated in parts of Central and Southern Europe (Fig. 1a). In central Germany as well as in parts of Greece and Spain unplanned canopy openings accounted for >50% of all area disturbed between 1986 and 2020. Conversely, planned canopy openings were most prevalent in Fennoscandia, the Baltic states, Poland, but also central Italy, where canopy openings from human activity accounted for >90% of all disturbances recorded (Fig. 1a).

### Disturbance change

In the early 21st century (i.e., the years 2001 to 2020), average disturbance area per year was 25% higher than in the late 20th century (1986 to 2000, Fig. 1b), corresponding to an average increase in the canopy area disturbed of 2626 km² per year. Both planned and unplanned canopy openings contributed to this increase: Planned canopy openings were on average 24% higher in the early 21st compared to the late 20th century, while the rates of unplanned canopy openings increased by 30% (Fig. 1b). As both types of disturbance increased significantly ($p < 0.05$ for both agents, non-parametric Van der Waerden test), the share of unplanned canopy openings on the total canopy area disturbed changed only marginally, increasing from 17% in the late 20th century to 18% in the early 21st century (Fig. 1c). In the year with the highest level of unplanned canopy opening (2000), ecological causes of tree mortality accounted for 35% of all area disturbed (Fig. 1c). Changes in unplanned canopy openings were spatially more variable than changes in planned canopy openings, with disturbances by ecological causes increasing by up to 1000% at the level of individual analysis units, while the maximum anthropogenic disturbance change remained an order of magnitude lower (Supplementary Fig. 1). Likewise, the increase in unplanned canopy openings was more variable in time than the increase in planned canopy openings. Disturbances from anthropogenic causes increased gradually and at a relatively steady rate throughout the 35-year observation period, while disturbances by ecological causes increased in pulses (Fig. 1b). The ten years with the highest overall area disturbed all occurred after 1999, peaking at 15,800 km² per year.

### Linked disturbances

Planned and unplanned canopy openings did not change independently of each other in Europe's forests, but their changes were linked. Specifically, we found that areas where planned canopy openings increased also experienced an increase in unplanned canopy openings,

a)

Unplanned canopy openings (%)

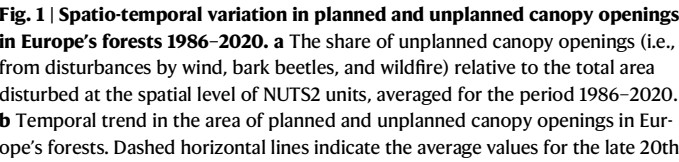

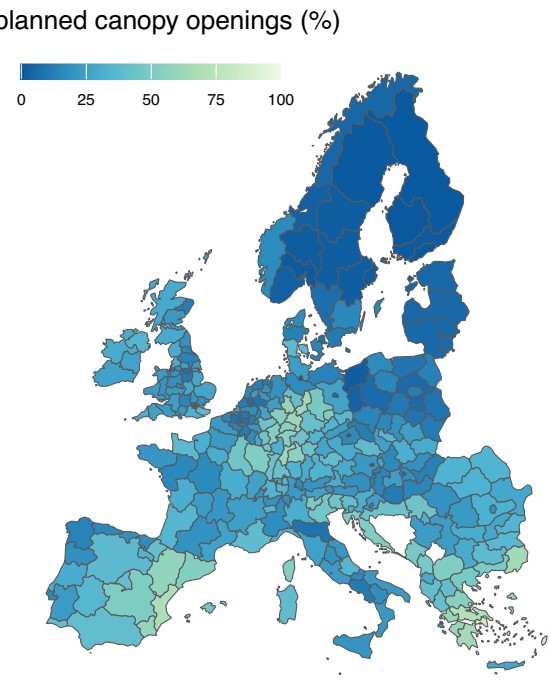

b)

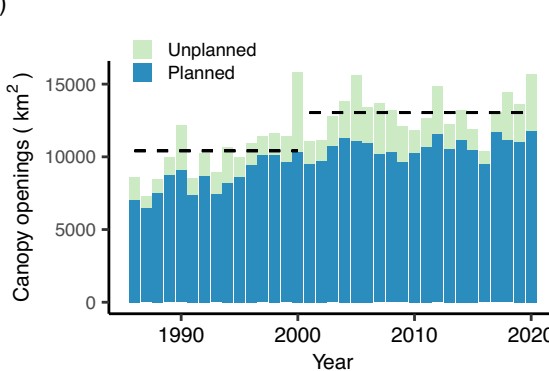

c)

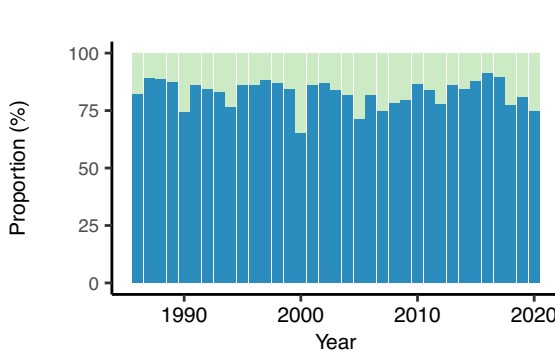

**Fig. 1 | Spatio-temporal variation in planned and unplanned canopy openings in Europe's forests 1986–2020. a** The share of unplanned canopy openings (i.e., from disturbances by wind, bark beetles, and wildfire) relative to the total area disturbed at the spatial level of NUTS2 units, averaged for the period 1986–2020. **b** Temporal trend in the area of planned and unplanned canopy openings in Europe's forests. Dashed horizontal lines indicate the average values for the late 20th and the early 21st century, respectively. **c** Relative share of planned and unplanned canopy openings on the overall area disturbed. NUTS is *Nomenclature des unités territoriales statistiques*, denoting administrative units nested within countries, used here as the main spatial analysis entities. Administrative boundaries © EuroGeographics.

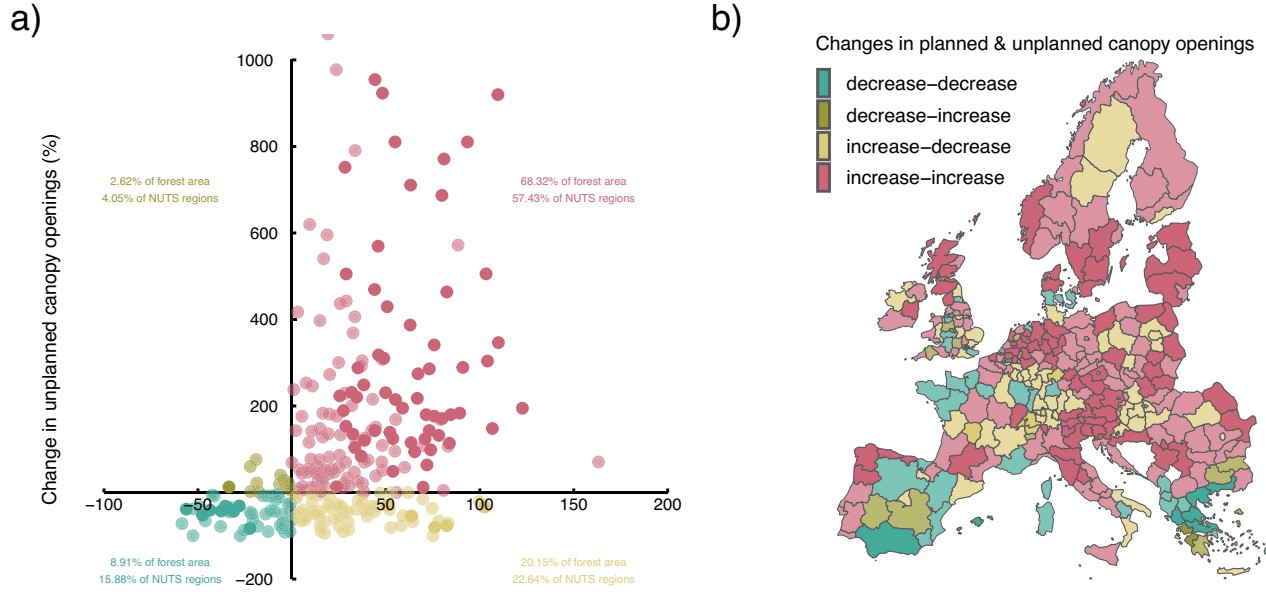

**Fig. 2 | Changes in planned and unplanned canopy openings are linked in Europe's forests. a** The increase in planned and unplanned canopy openings in Europe is not independent but linked, with a significant increase in disturbances caused by both anthropogenic and ecological causes in the majority of analysis units (57%), representing 68% of Europe's forest area. **b** Areas where both planned and unplanned canopy openings increase dominate in Central and Eastern Europe as well as southern Fennoscandia, while patterns are more variable in Western Europe and the Mediterranean. Shading differentiates significant changes ($p < 0.05$ in dark hues; non-parametric Van der Waerden test) from not significant ones (light hues). Reported changes contrast the early 21st century (2001–2020) to the late 20th century (1986–2000).

and vice versa (Fig. 2a). For 68% of the forest area (57% of analysis units), planned and unplanned canopy openings increased simultaneously (28% and 25% when only statistically significant changes at $p < 0.05$ were considered, using a non-parametric Van der Waerden test). In contrast, for only 9% of the forest area (16% of analysis units) the rates of both planned and unplanned canopy openings decreased. Increasing planned canopy openings co-occurred with decreasing unplanned canopy openings on 20% of the forest area, especially in regions that were affected by large storms in the late 20th century. For 3% of the forest area (6% of analysis units) planned canopy openings decreased while unplanned canopy openings increased, with only few of these changes being statistically significant. In Central and Eastern Europe as well as southern Fennoscandia, both planned and unplanned canopy openings increased (Fig. 2b). Patterns were more varied in Western Europe and the Mediterranean, where increases and decreases in both planned and unplanned canopy openings were observed. The positive link between planned and unplanned canopy openings was particularly strong for areas that had low to intermediate levels of unplanned canopy openings relative to the total area disturbed (cf. Fig. 1a).

## Discussion

We here show that changes in planned and unplanned canopy openings are linked in Europe's forests. Amplifying interactions between anthropogenic and ecological causes of disturbance can, for instance, arise from forest operations, such as cutting live trees to establish logging tracks (i.e., planned canopy opening) in order to create access for salvage logging following bark beetle attack (i.e., unplanned canopy opening)[14]. Likewise, planned canopy openings create edges (e.g., clearcut harvesting) or increase the surface roughness of the forest canopy (e.g., via thinning from above, retention forestry), which can increase the susceptibility to disturbances by agents such as windthrow and bark beetle outbreaks[15,16]. A further pathway for amplifying interactions is the cultivation of plantation forests (e.g., in

Southwestern Europe), which can intensify forest fire regimes[17]. However, increases between planned and unplanned canopy openings could also co-occur because they respond to the same drivers of global change and land-use history. Global change effects such as $CO_2$ fertilization and N deposition can—in combination with moderate warming —result in improved tree growth particularly in parts of Central and Northern Europe[18]. This increase in tree growth can increase both planned (e.g., increased harvesting frequency) and unplanned canopy openings (as taller trees are more susceptible to windthrow and more biomass means increased fuel for wildfire[19,20]). Furthermore, areas in which past land-use has favored highly productive but disturbance-prone tree species such as Norway spruce (*Picea abies* (L.) Karst.) often have high levels of planned canopy openings, because Norway spruce is the most important timber species on the European market. They are also particularly susceptible to disturbances from windthrow and bark beetles[19,21] and thus experience frequent unplanned canopy openings. Similar responses to global change and land-use history could thus lead to co-occurring changes in planned and unplanned canopy openings. We note that causal inference—whether unplanned canopy openings increase because of the increase in human land use and vice versa, or whether these changes simply co-occur—is challenging based on observational data[22], and should be the focus of future research.

Important implications arise from our finding that planned and unplanned canopy openings in Europe are not changing independently but do so as linked disturbances[23,24]. In general, our results underline that a coupled human and natural systems perspective[25,26] is needed to address the changing forest disturbance regimes of Europe in science and policy. Recognizing the causes of disturbance change as linked has a number of important implications: First, it highlights that disturbance change is not solely an effect of climate change (as is often portrayed in the public debate) but requires an integrated consideration of the ongoing changes in climate, forest structure, composition and land-use. Second, we document a co-occurring increase in planned and unplanned canopy openings in Europe's forests. This suggests that

maintaining and increasing forest health via forest risk management, aiming to increase the resistance to unplanned canopy openings and reduce their impacts through management[27,28], was not successfully able to counteract the unplanned canopy openings observed for large parts of Europe in the early 21st century. From this we conclude that the focus of forest policy and management should increasingly shift to coping with disturbances rather than aiming to prevent them, e.g., by fostering recovery from and resilience to disturbance[29,30]. Lastly, management should aim to break the link between changes in planned and unplanned canopy openings, by e.g., incorporating ecological agents of disturbance more integrally into forest planning and decision making. Reducing planned canopy openings—used in silviculture e.g., to regenerate forests and establish trees that are adapted to future climate conditions—and focusing regeneration and adaptation measures on areas affected by unplanned canopy openings could dampen the overall trend of increasing disturbance. Reducing planned canopy openings and managing for structural as well as compositional diversity in areas that experience a strong increase in unplanned canopy openings can have positive effects in the context of climate change mitigation, e.g., on the microclimatic buffering capacity and carbon storage potential of forests[31,32]. Utilizing unplanned canopy openings as opportunities for management would also align forest management interventions better with ecological processes (e.g., in terms of patch size distributions), a goal that many forest managers throughout Europe strive towards ("closer-to-nature forest management"[33]). Given that disturbances have profound implications on the services society derives from forests, we argue that the interactions between planned and unplanned canopy openings should be a central consideration in the stewardship of forest ecosystems in a changing world.

## Methods

Our analysis builds on an existing forest disturbance map for Europe (spatial grain: 30 m) for the years 1986–2020 that we developed previously[11]. The map indicates if and when high severity canopy openings occurred, with high severity canopy openings defined as major turnover in canopy trees (average canopy loss of 66 %, average recovery time to pre-disturbance canopy cover 30 years[34]). The map was created using supervised classification of spectral trajectories derived from satellite data (see ref. 11 for details). In order to attribute canopy openings to either being planned (i.e., caused by human land use) or unplanned (i.e., caused by wind, bark beetles and wildfire), we built on an attribution algorithm developed previously[12,13], but adapted it to the specific objectives of the current study. In essence, the algorithm first identifies individual disturbance patches using queen-contiguity, combining all pixels disturbed in the same year sharing either an edge or node. To account for temporal inaccuracies in the underlying disturbance map (e.g., satellite images were analyzed for the peak of the vegetation period, artificially splitting a fire that burns throughout the vegetation period into two fires occurring in consecutive years) all patches with a shared edge occurring in consecutive years were merged, assigning the disturbance year of the merged patch by a majority vote across all pixels. Subsequently, an exhaustive set of predictors was derived for each patch, including indicators describing the size and shape of the patch, the spectral characteristics before and during the opening of the canopy, and its landscape context, i.e., whether the patch occurred spatially and temporally clustered with other patches[13]. A full list of all predictors is given in ref. 12. Using this set of predictors, a random forest model was used to estimate the probability of each patch being an unplanned canopy opening. For training the random forest model, we built upon an existing reference database of 11,364 point occurrences of either fires and windthrows[12]. We extended this database to 12,571 point occurrences by adding information on recent windthrow events and bark beetle disturbances emerging since 2018[35]. Newly added data points were assessed following the same protocol as described in ref. 12, using

visual interpretation of canopy openings, satellite imagery, high-resolution imagery from Google Earth and existing databases to assign a point location to different causes of unplanned canopy openings (i.e., wind, bark beetles and wildfire). Wind events were verified with the help of the FORWIND database[36]. Fires were identified by cross-checking canopy openings with the EFFIS database (https://effis.jrc.ec.europa.eu), which is a pan-European database on forest fires. As guidance for visually interpreting bark beetle patches we used a set of papers describing recent outbreaks of bark beetles in Central Europe[37,38]. For the thus identified hotspots of recent bark beetle disturbance, we identified bark beetle disturbances by either red or gray standing conifer trees in high-resolution imagery, or by large clear-cut areas (i.e., orders of magnitude larger than those legally allowed in planned management interventions) indicating salvage logging of bark beetle-infested trees. Each point occurrence was linked to a canopy opening by unique id. In many cases, two or more occurrence points fell within the same disturbance patch (especially for large fires or windthrows), which reduced the number of patches with an agent label to 9256, including 6986 for windthrow, 727 for bark beetles and 1543 for wildfire.

Disturbances by wind, bark beetles and wildfire were jointly considered unplanned canopy openings. We note that we here focus on the root cause of tree mortality, i.e., an area initially affected by an ecological disturbance agent such as wind and subsequently salvage harvested was considered an unplanned canopy opening in our analysis, because the canopy opening would not have occurred in the observed form without the ecological disturbance agent. Drought often acts as predisposing factor for tree mortality from bark beetles and wildfire, but is not explicitly considered in our attribution model. Planned canopy openings mainly refer to timber harvesting, and only to a smaller extent to land use changes, as the latter account for only 2.6% of all canopy openings recorded across Europe[39]. As planned canopy openings dominate the disturbance regime of Europe[3,12,40], we did not specifically assign patches to planned canopy openings in our attribution, but rather used a random background sample to indicate the absence of unplanned canopy opening in our random forest model. Specifically, we randomly sampled 9256 patches (i.e., the same number as our unplanned sample) from all canopy openings in Europe that were not labeled unplanned and labeled them as planned canopy openings, resulting in a final reference database of 18,512 patches. The random forest model trained on this database was validated using spatial block 10-fold cross-validation with 0.5 km² hexagons used as folds, following suggestions in ref. 41. The cross-validated area under the receiver operating characteristic curve (AUC) was 0.98. To classify the probability of an unplanned canopy opening into the binary categories planned and unplanned, we optimized the probability cut-off using the F1-score derived from spatial cross-validation. The optimal threshold was 0.39, which resulted in an overall classification accuracy of 93%. We note that the cross-validated accuracies reported here do not represent map accuracies (i.e., a completely independent statistical estimation of map accuracies), which are difficult to estimate consistently for large-scale remote sensing datasets spanning several decades of data. To evaluate our results, we compared our estimates of unplanned canopy openings to a dataset on salvage logging provided by the Joint Research Center of the European Union[42]. This dataset contains information on the percentage of all fellings (in terms of timber volume) that come from salvage logging (i.e., unplanned logging operations triggered by ecological disturbances) for 16 European countries over 16 years (2004 to 2020). For comparison to this dataset, we aggregated the total area of unplanned canopy openings estimated here to per year and country values, and divided it by the total area of all canopy openings per year. Assuming that management responds to the large majority of unplanned canopy openings recorded in our dataset via salvage logging, we compared our rate of unplanned canopy openings to the rate of salvage logging from the database of

the Join Research Center of the European Union (Supplementary Fig. 2). We found a high correlation between both datasets at national level across all years (Pearson $r = 0.62$, Supplementary Fig. 2a) as well as at the level of individual years ($r = 0.58$, Supplementary Fig. 2b), supporting our map-based estimates of unplanned canopy openings. We note, however, that we did not expect a perfect correlation between the two datasets because of inherent differences in area-based estimates and timber volume-based estimates of canopy disturbance. Timber volume data do, for instance, include timber from thinning operations, which are non-stand replacing disturbances and thus not included in our disturbance map.

From the maps of planned and unplanned canopy openings we calculated annual map-based estimates of total canopy openings and the proportion of planned and unplanned canopy openings across Europe. All map-based estimates were calculated for canopy openings occurring in the late 20th century (i.e., between 1986 and 2000) and in the early 21th century (i.e., between 2001 and 2020). We chose a period comparison over a continuous trend analysis as our statistical approach as time series of total canopy openings are strongly driven by individual events (i.e., creating outliers in a statistical sense), making linear trend analyses inappropriate. We chose an arbitrary cutoff between the 20th and 21st century for our analysis, and included the year 2000 with the former period, as the major storm event "Lothar" which occurred in December 1999 is only recorded in the year 2000 in our data. Testing the sensitivity of our analysis to different cutoff years indicated similar results also for different cutoff years (Supplementary Fig. 3). We tested for differences between the two periods using a non-parametric Van Waerden test as implemented in the package 'coin'[43] in the R software and environment for statistical computing[44].

**Reporting summary**

Further information on research design is available in the Nature Portfolio Reporting Summary linked to this article.

## Data availability

The data of the European disturbance map Version 1.1.2 is deposited on Zenodo at: https://doi.org/10.5281/zenodo.3924380. The data used for the initial disturbance attribution algorithm is deposited on Zenodo at: https://zenodo.org/records/8202241. The data from the FORWIND database is deposited on figshare at: https://doi.org/10.6084/m9.figshare.9555008. All original data of this study are deposited on Zenodo at: https://doi.org/10.5281/zenodo.11070255.

## Code availability

The initial disturbance attribution algorithm is available under is deposited on Zenodo at: https://doi.org/10.5281/zenodo.4607164. All original code of this study is deposited on Zenodo at: https://doi.org/10.5281/zenodo.11070255.

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

## Acknowledgements

This work was supported by the European Union's Horizon 2020 research and innovation program under grant agreement no. 101000574 (RESONATE: Resilient forest value chains—enhancing resilience through natural and socio-economic responses). R.S. acknowledges further support from the European Research Council under the European Union's Horizon 2020 research and innovation program (Grant Agreement 101001905, FORWARD).

## Author contributions

Rupert Seidl: conceptualization, methodology, writing—original draft, project administration, funding acquisition. Cornelius Senf: conceptualization, methodology, software, validation, formal analysis, writing—review & editing, visualization.

## Funding

## Competing interests

The authors declare no competing interests.
