## [Peer Review File · Nature Communications]

Changes in planned and unplanned canopy openings are linked in Europe's forestsREVIEWER COMMENTS

Reviewer #1 (Remarks to the Author):

The authors used satellite data-based disturbance maps for Europe and machine learning to differentiate among disturbance agents to show changes in natural and human disturbances from 1986–2020 and at the level of NUTS2-regions. They found an increase in both natural and human disturbances over the considered period and that for over 71% of the forest area the increase of human and natural disturbances occurred simultaneously. They used the term “linked” to describe this co-occurrence. The novelty in this research is the identification of disturbance agents at large scales. The authors provide explanations for the linkage using a coherent theoretical model of causal interactions between human and natural disturbances. However, the data did not allow identifying the importance of individual interactions or whether the human and natural disturbance were linked by a common driver only. All in all, the study is very interesting, timely and novel both methodologically and results-wise. The manuscript is very easy to read, concise and well structured. I have only about 1.5 major comments and a few minor comments:

Looking at Figs 1 and 3b suggests that linked human and natural disturbances primarily occurred in areas with a low share of natural disturbances. You may want to mention this in the results section and discuss its implications. I could imagine that this would strengthen your argument for linkage due to management-induced disturbance susceptibility such as wind exposed edges. This prevalence of linkage in highly managed forests justifies somewhat that you primarily focus on (Norway) spruce dominated areas for the explanations on possible links between natural and human disturbances. However, since linked natural and human disturbances also seem prevalent in Portugal, south of France and large parts of Italy, I miss a discussion of the potential mechanisms causing linkages in these fire-dominated systems.

Minor comments

Line 30: Change “shorter” to smaller

Line 120: Something got lost. Sentence does not end properly.

Line 125: “Considerations” do not cause the observed interactions -> delete?

Line 156: You may want to mention that disturbance reduction management was SO FAR not successful. The reason could be that its effects play out in the long-term and are thus not yet observable and/or that it hasn't been implemented rigorously enough. You may want to mention this and then conclude that along with increased efforts for increasing resistance to disturbance, also the shift to coping is necessary.

Lines 165-168: You may want to mention increasing tree species and forest structural diversity as additional measures in the context of climate change mitigation (e.g., Mori et al. 2021: doi 10.1038/s41558-021-01062-1, Pedro et al. 2015: doi 10.1007/s00442-014-3150-0)

Reviewer #2 (Remarks to the Author):

(This is a joint review)

We appreciate the opportunity to review the manuscript. The study builds upon previous work by the authors using remote sensing and databases of forest disturbance to develop GIS-based maps of disturbance across Europe's forests. Creating a dataset of continental-scale disturbances and attributing their causes is a vast task. The results of the dataset have been widely applied by the authors and others. Thus the overall project of mapping forest disturbance is relevant and a valuable contribution to science. Another useful contribution is the joint consideration of types and causes of disturbance.

However, we have two broad concerns about the article, followed by several specific comments for the authors and editor to consider.

Broad concern 1: The article is essentially the same as citation #11, Senf and Seidl 2021, also in the Nature publication group. That article reported on European forest disturbance mapping through 2017. The current version updates the dataset to 2020. It is not clear that there is significant new scientific value in publishing similar datasets a few years apart.

Broad concern 2: The article contrasts "human" and "natural" disturbances. These categories are defined in the first paragraph (lines 22-34). Later in the Methods, the authors state that "our distinction between natural and human agents is agnostic to the fact that humans are changing natural disturbance regimes, e.g. via anthropogenic climate change or the human ignition of fires. Consequently, we here jointly address fire, wind and bark beetles as natural disturbances" (L 285-287). While it may be the authors' opinion that that the categories of "human" and "natural" can be segregated, this distinction seems complex and poorly supported to us. Our perspective, as scientists who work on forest disturbances in North America, the Mediterranean, and South Africa, leads us to push back. A human-caused fire in a *Pinus nigra* plantation in Spain is "natural"? Windthrow or bark beetles in a *Pseudotsuga* plantation in Central Europe are "natural" disturbances? The forests of Europe may be among the globe's least likely places to assert without justification that "human" and "natural" disturbances are distinctly different things, given the long history of forest use by dense human populations. This issue raises a further concern with respect to this manuscript because its premise is based on comparing "human" vs. "natural" disturbances. We also suggest providing additional context for acknowledging spatial and temporal variation in disturbances across the broader region. Factors such as geographical features, land-use changes, and/or forest management practices contribute to observed patterns. In our view, while there are meaningful distinctions that could be made between various disturbances, it is a complex and subtle situation that should not be glossed over as in the present text.

Specific issues:

1. Title: The title refers to "Europe's forests" but the actual area addressed corresponds to the European Union and partners + UK, which comprise about 40% of the area of the European continent. It would be appropriate to revise the title. We understand that it is common to refer to the EU as "Europe" but it is imprecise and exclusionary, just as when authors refer to "North America" when they mean the USA + Canada.
2. L. 50-51: Authors state that human land use strongly dominates the disturbance regime in Europe, however, the authors indicate in the Methods that "Human disturbance mainly refers to timber-harvesting, as land-cover changes...only account for 2.6% of all canopy disturbances recorded in Europe". This should be clearer or outlined in the main text, as is otherwise misleading to readers.
3. L. 57-58: One example of the result of the confusion between "human" and "natural" disturbances is shown here, since almost certainly the predominant "natural" disturbances in Greece and Spain were human-ignited fires in human-created fuelbeds (tree plantations and abandoned agricultural lands), while the "natural" disturbances in Germany were likely entirely different (perhaps insect and storm damage?). The value to scientists and managers of untangling the different causes of disturbance is not realized when the causes are conflated.
4. L. 70-87: It would be helpful to provide more context about the disturbance statistics discussed here and in Fig. 2. While the disturbance rates were statistically significantly different and changes in disturbance in terms of percentages reached large amounts (25-33%), the actual magnitudes of the rates were not necessarily greatly different, especially for natural disturbances (0.08% vs. 0.10%). Fig. 2b seem to show a plateau in disturbance after 2000 or 2002. Thus the text in this section may come across as alarmist.
5. L. 70-87: While not explicitly stated in the text, it is evident that much of the human-caused disturbance described in this study results from commercial forest management in Scandinavia and northern Europe. These are productive forest regions where timber is grown on rotations that can match the temporal scale of the present study (35 years). Therefore, while many areas of forest will have been disturbed by timber harvest during the study period, they would also have been

regenerated by planting or natural regeneration. Examining the trend over time solely in the context of disturbance is therefore misleading, because it conflates sustainable forest management with, for example, long-term loss of forest due to severe wildfire, as is also occurring in other areas of Europe. The lack of recognition of reforestation could also contribute to alarmist interpretations of the study's findings, as noted above. The Hansen Global Forest Change dataset (https://developers.google.com/earth-engine/datasets/catalog/UMD_hansen_global_forest_change_2022_v1_10), while having its own challenges, does incorporate forest gain as well as loss, providing a more balanced picture.

6. L. 120: Incomplete sentence.

7. L. 132-136: Increased tree growth due to the factors mentioned is occurring at a number of places around the world, especially where cold temperatures limited past growth as in the Central European sites cited, but the opposite is expected in warm, semi-arid regions such as Mediterranean Europe. The Pausas and Ribeiro (2013) article cited in fact makes this distinction, but the authors have overlooked it. This section should be rewritten to address the diversity of European forests rather than making a blanket statement that tree growth will increase.

8. L. 141-145: It is not clear why the authors assert that "causal inference ... is not possible with our data." Perhaps they mean that assigning cause and effect cannot be done conclusively with observational data, which may be technically true, but estimations of causal factors and their strengths is the basis of vast areas of science. It seems rather that they chose to simplify this study by choosing not to investigate the interactions between disturbances and a variety of other data sets such as climate, vegetation, population, economic, and other available data sources.

9. L. 145-153: Strongly agree that the inherent complexity of interlinked disturbances is essential to consider in adaptive management under climate and other pressures. This message is critical but highlights the frustrating aspect of this manuscript in that the article itself makes numerous oversimplifications, as noted throughout this review.

10. L. 154-157: The authors refer to "forest risk management" as if it were an adopted universal policy that should have brought change by the early 21st century. In fact, the citation (#26) is to an academic article about risk assessment, not an established forest management policy. If an active policy did not exist, it is not surprising that management "... was not successfully able to counteract increases in natural disturbances."

11. L. 161 and elsewhere: A comma should be used after "e.g."

12. L. 162-168: The recommendation that forests should be managed for continuous canopy cover is not reasonable. A continuous canopy may have certain properties that could be considered useful for some management goals, such as reducing the likelihood of windthrow of trees or maintaining a shaded understory. On the other hand, a continuous canopy may have certain properties that could be considered detrimental for other management goals, such as providing continuous fuel for a megafire. It is not appropriate in a scientific article to make anecdotal, generalized recommendations without clarifying the complex issues involved.

13. L. 252, 257: As the authors are the ones who created the maps that the Methods refer to, it would be appropriate to make this known to the reader. For example, instead of saying "an existing forest map", it would be appropriate to say "an existing forest map that we developed".

14. L. 288-290: The choice to exclude drought is related to the comment about avoiding analysis of the data. As noted above, it is not that it is "not possible" to assess the factors associated with disturbance, but rather that the authors chose not to do so. This should be stated in the main text, not Methods.

15. L. 295: It is not clear why the authors grouped wind and bark beetle disturbances together, especially as they described in detail (half of the Methods section) the different data sources that allowed them to assign these disturbances as distinct. It may be true that "they frequently occur together in Europe's forests" but it seems logical that they often do not co-occur, as is observed in the interacting beetle-fire disturbances that have been observed in many pine forests (e.g., Canadian Rockies). It is challenging to the reader to see that (1) the authors claim to have identified wind and beetle disturbances separately and accurately, (2) the authors hold the data to assess quantitatively the degree to which wind and beetle disturbance co-occur, but (3) the authors choose not to use their data for this analysis and instead lump the data with an anecdotal remark about co-occurrence. It also

raises the question of whether the authors did this to arrive at lower error estimates (L. 298-300)?

16. L. 300-303: The authors state that they cannot determine map accuracy because it is "difficult to estimate consistently." No doubt it is difficult, but is it appropriate to publish numerous studies without attempting to verify the data, at least with certain case studies? It would seem logical that there could be many independent assessments that could be compared with the current study, such as the Hansen data set mentioned above or case studies of fires in Greece or Spain.

17. L. 307: The definition of "Europe" should be presented and justified early in the text, not at the end.

18. Extended Data Fig. 2: This figure presents graphical flowcharts of the processes described in the text on L. L. 162-168 (panel a) and L. 132-136 (panel b). As we commented above, those processes are highly oversimplified in the text (i.e., managing for continuous forest cover is not a universally appropriate strategy, global change does not universally increase forest growth). Therefore, we recommend removing this figure.

19. References: We note that 14 out of 39 references (36%) are self-citations of articles co-authored by the present authors. That appears excessive, especially as many of these cited studies draw on the same data set as in this manuscript to carry out similar analyses. To address issues such as map accuracy, noted above, the authors would better serve the reader by using and citing independent analyses of disturbance and forest change.

Reviewer #3 (Remarks to the Author):

I participated in the manuscript review alongside one of the reviewers who supplied the listed reports. This engagement is part of the Nature Communications initiative, designed to support training in peer review and recognize the contributions of Early Career Researchers involved in co-reviewing manuscripts.

Reviewer #1 (Remarks to the Author):

The authors used satellite data-based disturbance maps for Europe and machine learning to differentiate among disturbance agents to show changes in natural and human disturbances from 1986–2020 and at the level of NUTS2-regions. They found an increase in both natural and human disturbances over the considered period and that for over 71% of the forest area the increase of human and natural disturbances occurred simultaneously. They used the term “linked” to describe this co-occurrence. The novelty in this research is the identification of disturbance agents at large scales. The authors provide explanations for the linkage using a coherent theoretical model of causal interactions between human and natural disturbances. However, the data did not allow identifying the importance of individual interactions or whether the human and natural disturbance were linked by a common driver only. All in all, the study is very interesting, timely and novel both methodologically and results-wise. The manuscript is very easy to read, concise and well structured. I have only about 1.5 major comments and a few minor comments.

Response: We thank the Reviewer for their positive assessment of our work, and for the helpful comments on how to improve it further!

Looking at Figs 1 and 3b suggests that linked human and natural disturbances primarily occurred in areas with a low share of natural disturbances. You may want to mention this in the results section and discuss its implications. I could imagine that this would strengthen your argument for linkage due to management-induced disturbance susceptibility such as wind exposed edges. This prevalence of linkage in highly managed forests justifies somewhat that you primarily focus on (Norway) spruce dominated areas for the explanations on possible links between natural and human disturbances.

Response: Thank you for this observation, this is indeed a relevant and interesting detail of our results that deserves explicit mentioning in the text. We have now added a statement highlighting this aspect for the reader (lines 115-117).

However, since linked natural and human disturbances also seem prevalent in Portugal, south of France and large parts of Italy, I miss a discussion of the potential mechanisms causing linkages in these fire-dominated systems.

Response: Good point! One potential pathway is the extensive cultivation of (highly flammable) plantations in south-western Europe (mostly Pinus and Eucalyptus species), which could in turn amplify interactions between planned canopy openings and unplanned openings caused by wildfire. We have now added this potential interaction explicitly to the discussion in lines 138-140.

Minor comments

Line 30: Change “shorter” to smaller

Response: Done!

Line 120: Something got lost. Sentence does not end properly.

Response: Figure caption revised – thanks for spotting this!

Line 125: “Considerations” do not cause the observed interactions -> delete?

Response: Revised for clarity. “Considerations” is now omitted as suggested.

Line 156: You may want to mention that disturbance reduction management was SOFAR not successful. The reason could be that its effects play out in the long-term and are thus not yet observable and/or that it hasn't been implemented rigorously enough. You may want to mention this and then conclude that along with increased efforts for increasing resistance to disturbance, also the shift to coping is necessary.

Response: We agree that there is a temporal element to this. However, our sentence clearly states that we here discuss the increase in the early 21st century, and this was clearly not prevented by risk management (which is ongoing in Europe's forestry at least since the 1950s). We have thus retained the sentence as is.

Lines 165-168: You may want to mention increasing tree species and forest structural diversity as additional measures in the context of climate change mitigation (e.g., Mori et al. 2021: doi 10.1038/s41558-021-01062-1, Pedro et al. 2015: doi 10.1007/s00442-014-3150-0)

Response: Fully agreed! We've revised this sentence to explicitly mention structural and compositional diversity as potential management strategies for coping with disturbance change. However, we've not added the suggested references, as we were involved in both studies, which would further increase the self-citation rate that Reviewer #2 found already to be high.

Reviewer #2 (Remarks to the Author):

(This is a joint review)

We appreciate the opportunity to review the manuscript. The study builds upon previous work by the authors using remote sensing and databases of forest disturbance to develop GIS-based maps of disturbance across Europe's forests. Creating a dataset of continental-scale disturbances and attributing their causes is a vast task. The results of the dataset have been widely applied by the authors and others. Thus the overall project of mapping forest disturbance is relevant and a valuable contribution to science. Another useful contribution is the joint consideration of types and causes of disturbance.

Response: Thank you for this overall positive assessment of our work, and also for the many helpful suggestions to improve it further!

However, we have two broad concerns about the article, followed by several specific comments for the authors and editor to consider. Broad concern 1: The article is essentially the same as citation #11, Senf and Seidl 2021, also in the Nature publication group. That article reported on European forest disturbance mapping through 2017. The current version updates the dataset to 2020. It is not clear that there is significant new scientific value in publishing similar datasets a few years apart.

Response: We have to disagree strongly here, as the work presented here is novel and has not yet been presented before. It is correct that we build upon our previously published forest disturbance map – and hence refer to it in the text. However, the previous publication highlighted by the Reviewer did not include an attribution of disturbance patches to different disturbance agents, and hence did also not focus on their interactions, which were our two main objectives in the current work. As there is an ongoing discussion in Europe about the causes of forest disturbance change (e.g., e.g., Ceccherini et al. 2021, Nature 583, 72-77, Palahi et al. 2022, Nature 592, E15-E17, Wernick et al. 2022, Nature 592, E13-E14, Ceccherini et al. 2022, Nature 592, E18-E23), our work is highly novel and fulfils an important information need for European forestry. This value was also highlighted by Reviewer #1, and we are confident that our novel results and findings will be received with high interest by the community.

Broad concern 2: The article contrasts “human” and “natural” disturbances. These categories are defined in the first paragraph (lines 22-34). Later in the Methods, the authors state that “our distinction between natural and human agents is agnostic to the fact that humans are changing natural disturbance regimes, e.g. via anthropogenic climate change or the human ignition of fires. Consequently, we here jointly address fire, wind and bark beetles as natural disturbances” (L 285-

287). While it may be the authors' opinion that that the categories of "human" and "natural" can be segregated, this distinction seems complex and poorly supported to us. Our perspective, as scientists who work on forest disturbances in North America, the Mediterranean, and South Africa, leads us to push back. A human-caused fire in a *Pinus nigra* plantation in Spain is "natural"? Windthrow or bark beetles in a *Pseudotsuga* plantation in Central Europe are "natural" disturbances? The forests of Europe may be among the globe's least likely places to assert without justification that "human" and "natural" disturbances are distinctly different things, given the long history of forest use by dense human populations. This issue raises a further concern with respect to this manuscript because its premise is based on comparing "human" vs. "natural" disturbances. We also suggest providing additional context for acknowledging spatial and temporal variation in disturbances across the broader region. Factors such as geographical features, land-use changes, and/or forest management practices contribute to observed patterns. In our view, while there are meaningful distinctions that could be made between various disturbances, it is a complex and subtle situation that should not be glossed over as in the present text.

Response: This is a very good point, and we fully agree that calling something "natural" in the context of Europe's forests is problematic, to say the least. We had not reflected much about this previously, simply because the term "natural disturbance" is widely used in the community to describe canopy openings by ecological agents such as wildfire, wind and bark beetles (cf. Patacca et al. 2022, GCB 10.1111/gcb.16531). However, we agree that this is not a good argument for perpetuating a terminology... We hence went back to the drawing board, as a number of other comments of the Reviewers and Editor (e.g., on independent evaluation) also required us to make changes to the attribution of disturbances done here. Ultimately, the distinction that we make is between canopy openings that are planned and those that are unplanned. So what we actually compare are changes in scheduled management interventions and in stochastic, unplanned canopy openings from ecological agents. We have reworked our entire attribution modeling, redone all our analyses and display items, and revised the text to incorporate this new attribution of disturbance types into our manuscript. We have omitted the term "natural" entirely in the revised text, in order to avoid confusion of the reader. This revision has also allowed us to address a number of other comments raised by the Reviewer (see below). But specifically, it has increased the clarity of our work and findings, and has reduced the ambiguity pointed out by the Reviewer regarding the terminology used previously. Thanks for the suggestion, and for making us rethink what it is we are actually doing here!

Specific issues:

1. Title: The title refers to "Europe's forests" but the actual area addressed corresponds to the European Union and partners + UK, which comprise about 40% of the area of the European continent. It would be appropriate to revise the title. We understand that it is common to refer to the EU as "Europe" but it is imprecise and exclusionary, just as when authors refer to "North America" when they mean the USA + Canada.

Response: This is a good point! We thought hard about this, particularly from the perspective of conveying to our readership what the spatial focus of our work is (this being the title of the manuscript, it is of utmost importance to attract the attention of readers who might be interested in our work). However, we did not find a better term than Europe's forests, as neither talking about the European Union nor some other political construct would be correct. But we agree that this issue is important, which is why we now explicitly detail our geographic study area for the reader already in the introduction (lines 55-58). We have also extended the previous text to make clear that we focus

on all countries predominately situated in continental Europe, which excludes countries such as Russia (ca. 25% of its area in Europe), Turkey (ca. 3% of its area in Europe) and Kazakhstan (ca. 5% of its area in Europe). We also note that we deliberately do not refer to the forests of Europe, but rather use “Europe’s forests”. This may sound as a small semantic detail, but it underlines that all of the forests we analyze are located in Europe, but we do not make the claim of analyzing the entirety of Europe’s forests. Thus, after careful consideration, and in the absence of a better alternative, we have retained the original formulation.

2. L. 50-51: Authors state that human land use strongly dominates the disturbance regime in Europe, however, the authors indicate in the Methods that “Human disturbance mainly refers to timber-harvesting, as land-cover changes...only account for 2.6% of all canopy disturbances recorded in Europe”. This should be clearer or outlined in the main text, as is otherwise misleading to readers.

Response: We agree that this was confusing, as we wrongfully wrote land cover changes, while talking about land use changes. This is rectified in the revised version of the manuscript.

3. L. 57-58: One example of the result of the confusion between “human” and “natural” disturbances is shown here, since almost certainly the predominant “natural” disturbances in Greece and Spain were human-ignited fires in human-created fuelbeds (tree plantations and abandoned agricultural lands), while the “natural” disturbances in Germany were likely entirely different (perhaps insect and storm damage?). The value to scientists and managers of untangling the different causes of disturbance is not realized when the causes are conflated.

Response: Agreed! As described above, we have rethought our objective, redone our analyses and rewritten our paper. The distinction between human and natural is now omitted here (and in other instances), in order to increase the utility of our work for the community.

4. L. 70-87: It would be helpful to provide more context about the disturbance statistics discussed here and in Fig. 2. While the disturbance rates were statistically significantly different and changes in disturbance in terms of percentages reached large amounts (25-33%), the actual magnitudes of the rates were not necessarily greatly different, especially for natural disturbances (0.08% vs. 0.10%). Fig. 2b seem to show a plateau in disturbance after 2000 or 2002. Thus the text in this section may come across as alarmist.

Response: We have revised the text as well as redone Figure 2 in light of the new analyses and scope of the manuscript.

5. L. 70-87: While not explicitly stated in the text, it is evident that much of the human-caused disturbance described in this study results from commercial forest management in Scandinavia and northern Europe. These are productive forest regions where timber is grown on rotations that can match the temporal scale of the present study (35 years). Therefore, while many areas of forest will have been disturbed by timber harvest during the study period, they would also have been regenerated by planting or natural regeneration. Examining the trend over time solely in the context of disturbance is therefore misleading, because it conflates sustainable forest management with, for example, long-term loss of forest due to severe wildfire, as is also occurring in other areas of Europe. The lack of recognition of reforestation could also contribute to alarmist interpretations of the study’s findings, as noted above. The Hansen Global Forest Change dataset (<https://developers.google.com/earth->

engine/datasets/catalog/UMD_hansen_global_forest_change_2022_v1_10), while having its own challenges, does incorporate forest gain as well as loss, providing a more balanced picture

Response: As suggested by the Reviewer we have thoroughly revised our analyses and reframed our interpretation. We are now analyzing planned vs. unplanned canopy openings, and a high share of planned canopy openings e.g. in Fennoscandia clearly reflects the processes noted by the Reviewers. This is also highlighted for the reader in lines 70-73. As for the inclusion of regeneration/ regrowth/ recovery, we note that we have done this in a previous paper (Senf and Seidl 2022, GEB, 10.1111/geb.13406), while we here focus on disturbance interactions, an element hitherto unexplored at the continental scale for Europe's forests.

6. L. 120: Incomplete sentence.

Response: Revised!

7. L. 132-136: Increased tree growth due to the factors mentioned is occurring at a number of places around the world, especially where cold temperatures limited past growth as in the Central European sites cited, but the opposite is expected in warm, semi-arid regions such as Mediterranean Europe. The Pausas and Ribeiro (2013) article cited in fact makes this distinction, but the authors have overlooked it. This section should be rewritten to address the diversity of European forests rather than making a blanket statement that tree growth will increase.

Response: Good point, we've made the sentence more specific to avoid too general blanket statements.

8. L. 141-145: It is not clear why the authors assert that "causal inference ... is not possible with our data." Perhaps they mean that assigning cause and effect cannot be done conclusively with observational data, which may be technically true, but estimations of causal factors and their strengths is the basis of vast areas of science. It seems rather that they chose to simplify this study by choosing not to investigate the interactions between disturbances and a variety of other data sets such as climate, vegetation, population, economic, and other available data sources.

Response: We agree that our original statement was worded too strongly. We have now revised the statement to make more clear that such causal analyses are challenging (but still entirely possible, as correctly pointed out by the Reviewer). However, there are broader conceptual issues beyond including a variety of datasets, as discussed in depth in Ferraro et al. (2019, PNAS 10.1073/pnas.1805563115), which is the reference given in the text.

9. L. 145-153: Strongly agree that the inherent complexity of interlinked disturbances is essential to consider in adaptive management under climate and other pressures. This message is critical but highlights the frustrating aspect of this manuscript in that the article itself makes numerous oversimplifications, as noted throughout this review.

Response: Thank you for this comment, we are happy that we here highlight an important issue in our discussion. We feel that through addressing the comments of both Reviewers throughout the text we've also improved our manuscript further and rectified the issues mentioned by the Reviewer here.

10. L. 154-157: The authors refer to “forest risk management” as if it were an adopted universal policy that should have brought change by the early 21st century. In fact, the citation (#26) is to an academic article about risk assessment, not an established forest management policy. If an active policy did not exist, it is not surprising that management “... was not successfully able to counteract increases in natural disturbances.”

Response: There is indeed no universal risk management throughout Europe. However, there is widespread political agreement on “maintaining forest health”, which in this context means to prevent unplanned forest canopy openings. This is agreed by all forestry ministers of the European Union, and is codified into the pan-European indicators of sustainable forest management. We have now revised the statement to make this more clear, and have also added another reference to said pan-European indicators.

11. L. 161 and elsewhere: A comma should be used after “e.g.”.

Response: Thanks! Now consistently implemented throughout the text.

12. L. 162-168: The recommendation that forests should be managed for continuous canopy cover is not reasonable. A continuous canopy may have certain properties that could be considered useful for some management goals, such as reducing the likelihood of windthrow of trees or maintaining a shaded understory. On the other hand, a continuous canopy may have certain properties that could be considered detrimental for other management goals, such as providing continuous fuel for a megafire. It is not appropriate in a scientific article to make anecdotal, generalized recommendations without clarifying the complex issues involved.

Response: We agree that our original formulation was not very clear and potentially misleading. We have revised the sentence, taking up a suggestion made by Reviewer #1, and now refer to increasing the structural and compositional complexity of forest ecosystems.

13. L. 252, 257: As the authors are the ones who created the maps that the Methods refer to, it would be appropriate to make this known to the reader. For example, instead of saying “an existing forest map”, it would be appropriate to say “an existing forest map that we developed”.

Response: Changed as suggested.

14. L. 288-290: The choice to exclude drought is related to the comment about avoiding analysis of the data. As noted above, it is not that it is “not possible” to assess the factors associated with disturbance, but rather that the authors chose not to do so. This should be stated in the main text, not Methods.

Response: Section revised to reflect the new focus of the study, comment no longer 100% applicable due to the changed focus of the analysis.

15. L. 295: It is not clear why the authors grouped wind and bark beetle disturbances together, especially as they described in detail (half of the Methods section) the different data sources that allowed them to assign these disturbances as distinct. It may be true that “they frequently occur together in Europe’s forests” but it seems logical that they often do not co-occur, as is observed in

the interacting beetle-fire disturbances that have been observed in many pine forests (e.g., Canadian Rockies). It is challenging to the reader to see that (1) the authors claim to have identified wind and beetle disturbances separately and accurately, (2) the authors hold the data to assess quantitatively the degree to which wind and beetle disturbance co-occur, but (3) the authors choose not to use their data for this analysis and instead lump the data with an anecdotal remark about co-occurrence. It also raises the question of whether the authors did this to arrive at lower error estimates (L. 298-300)?

Response: We agree that this grouping was not fully clear in the original submission. As suggested by the Reviewer, we have now reconsidered our categories and revised our approach. As the revised version of the manuscript now distinguishes between planned and unplanned canopy openings, the initial issue highlighted by the Reviewer is no longer applicable.

16. L. 300-303: The authors state that they cannot determine map accuracy because it is “difficult to estimate consistently.” No doubt it is difficult, but is it appropriate to publish numerous studies without attempting to verify the data, at least with certain case studies? It would seem logical that there could be many independent assessments that could be compared with the current study, such as the Hansen data set mentioned above or case studies of fires in Greece or Spain.

Response: We agree that evaluation of our results is crucial. We, however, do not think that the Hansen dataset would be a good candidate for evaluating our approach, as the Hansen data only spans part of the time series analyzed here (starting in 2000), as both datasets are based on Landsat data (and are thus not independent of each other), and, most importantly, as the Hansen dataset does not distinguish between different disturbance agents. However, we agree on the importance of this aspect, and have explored a number of options to better illustrate the validity of our approach for the reader. In this regard, the revised focus of our study on planned vs. unplanned forests has opened up new avenues, as datasets on salvage logging (representing timber extraction after an unplanned canopy opening) exist and can be explored. Specifically, we have conducted a new evaluation exercise, by comparing our newly estimated share of unplanned canopy openings (derived of area of unplanned canopy openings divided by all canopy openings * 100) to the share of timber volume from salvage harvest (derived as salvaged timber divided by total timber harvest * 100) for 16 European countries and 16 years (data from the Joint Research Center of the European Union <https://data.jrc.ec.europa.eu/dataset/2100b612-a4b0-4897-829b-72b7b1e5782c>). This allows us to both evaluate geographical differences (between countries) as well as the differences between years, which are now both shown in the newly added Extended Data Figure 3. The results show high correspondence, with Pearson's r of between 0.58 and 0.62. We thank the Reviewers for pushing us towards digging deeper into the evaluation of our data, as this newly added analysis has further improved the value of our contribution.

17. L. 307: The definition of “Europe” should be presented and justified early in the text, not at the end.

Response: Done!

18. Extended Data Fig. 2: This figure presents graphical flowcharts of the processes described in the text on L. L. 162-168 (panel a) and L. 132-136 (panel b). As we commented above, those processes are highly oversimplified in the text (i.e., managing for continuous forest cover is not a universally appropriate strategy, global change does not universally increase forest growth). Therefore, we recommend removing this figure.

Response: Done!

19. References: We note that 14 out of 39 references (36%) are self-citations of articles co-authored by the present authors. That appears excessive, especially as many of these cited studies draw on the same data set as in this manuscript to carry out similar analyses. To address issues such as map accuracy, noted above, the authors would better serve the reader by using and citing independent analyses of disturbance and forest change.

Response: We have now reduced the amount of self citations as suggested by substituting other references in a total of four instances (12% of cases). However, we also would like to point out that our work strongly builds on previous efforts (as also noted by the Reviewer), and citing them is vital for the reader to understand how the current work dovetails with previous efforts. With regard to the accuracy of our estimates we've taken up the suggestion of the Reviewer and have conducted additional analyses, comparing our estimates against independent data (see detailed comment above).

Reviewer #3 (Remarks to the Author):

I participated in the manuscript review alongside one of the reviewers who supplied the listed reports. This engagement is part of the Nature Communications initiative, designed to support training in peer review and recognize the contributions of Early Career Researchers involved in co-reviewing manuscripts.

Response: Thank you for your efforts and constructive comments!

REVIEWERS' COMMENTS

Reviewer #1 (Remarks to the Author):

I had few concerns to the original submission. The authors fully addressed them.

Reviewer #1 (Remarks on code availability):

I read through the R code. The code cannot be run, because the data the code reads in is not available. The authors state that the data will be made available once the manuscript is published. The code looked clean and easy to understand. However, it distinguished between human and natural disturbances, whereas in the revised manuscript its planned and unplanned disturbances. This revision should also be reflected in the published code to avoid confusion.

Reviewer #2 (Remarks to the Author):

(This is a joint review)

The authors have provided useful responses to the original review comments. Overall, the manuscript is substantially improved. We appreciate their work. We have only one significant remaining concern, item #6 in the list below.

Specific comments:

1. The authors addressed our original concern that the work did not differ sufficiently from their previous publication. We appreciate their explanation.
2. The authors also undertook a substantial revision to move from the "human/natural" categorization to "unplanned/planned" terminology. This change addressed our concerns and represents a more useful and generalizable perspective, thank you.
3. Other useful changes included better definition of "Europe," clarification of a number of terms and concepts, changes to the references, and adding the analysis of salvage logging now shown in Extended Data Figure 3.
4. Line 101-102: The text states that planned and unplanned canopy openings did not change independently of each other, suggesting a strong linkage. However, this contrasts with the subsequent mention (line 109-111) of areas where planned canopy openings decreased while unplanned canopy openings increased, indicating some degree of independence. It may be useful to mention some of the specific places with reasons as to why they may have showed this divergent response?
5. Implications on Forest Management: The text suggests a shift in forest management towards embracing disturbances rather than solely preventing them, recognizing their natural role in ecosystems (line 169-171). It also stresses the importance of including ecological agents in planning (line 173-174), implying a more proactive approach. However, these perspectives seem inconsistent. To reconcile them, the text could propose integrating reactive and proactive elements, like adaptive management, to respond to disturbances while enhancing resilience for the future.
6. One issue was not satisfactorily resolved: regeneration (comment #5). We pointed out that avoiding discussion of regeneration could be misleading because it conflates sustainable forest management with long-term loss of forest. The authors responded that they addressed regeneration/regrowth/ recovery in a previous paper while the present one focuses on disturbance. In our view, that response does not address the fact that some disturbances have short-term effects with minimal impact (e.g., harvest of a plantation followed by replanting) while others have long-term, high-impact effects (e.g., deforestation resulting from severe disturbance such as fire followed by regeneration failure). This could be dealt with in the manuscript by (1) acknowledging that different post-disturbance trajectories have different effects, and (2) using a snapshot of the data from the

previous study, give the reader a general idea of the degree to which regeneration/regrowth/ recovery is occurring. For example, "overall, X% of canopy openings in Europe return to forest cover within 15 years" or whatever the case might be. Additionally, for forest management implications, highlighting or acknowledging management practices aimed at promoting regeneration, such as afforestation, reforestation and silvicultural techniques, is essential. This will help demonstrate how sustainable forest management strategies can mitigate long-term loss for forest cover resulting from severe disturbances (see related comment above on implications to management too).

Reviewer #3 (Remarks to the Author):

Reviewer #1

I had few concerns to the original submission. The authors fully addressed them.

Response: Thank you for your help in improving our work further!

I read through the R code. The code cannot be run, because the data the code reads in is not available. The authors state that the data will be made available once the manuscript is published. The code looked clean and easy to understand. However, it distinguished between human and natural disturbances, whereas in the revised manuscript it planned and unplanned disturbances. This revision should also be reflected in the published code to avoid confusion.

Response: Thank you for spotting this! We have now revised the terminology used in the code to be in line with the terms used in the manuscript text. We also updated all code and data, and it is now possible to rerun all analysis with the code and data supplied under:
<https://doi.org/10.5281/zenodo.11070255>

Reviewer #2

The authors have provided useful responses to the original review comments. Overall, the manuscript is substantially improved. We appreciate their work. We have only one significant remaining concern, item #6 in the list below.

Response: Thank you for your valuable comments on the earlier version of our manuscript! For responses to the remaining comments see below.

1. The authors addressed our original concern that the work did not differ sufficiently from their previous publication. We appreciate their explanation.

Response: Thank you!

2. The authors also undertook a substantial revision to move from the “human/natural” categorization to “unplanned/planned” terminology. This change addressed our concerns and represents a more useful and generalizable perspective, thank you.

Response: Thanks for making us reconsider our initial approach, we feel that the revisions have considerably improved our work further!

3. Other useful changes included better definition of “Europe,” clarification of a number of terms and concepts, changes to the references, and adding the analysis of salvage logging now shown in Extended Data Figure 3.

Response: Thank you!

4. Line 101-102: The text states that planned and unplanned canopy openings did not change independently of each other, suggesting a strong linkage. However, this contrasts with the subsequent mention (line 109-111) of areas where planned canopy openings decreased while unplanned canopy openings increased, indicating some degree of independence. It may be useful to mention some of the specific places with reasons as to why they may have showed this divergent response?

Response: We have now added another line to the text, specifically referring to the case of increasing planned canopy openings and decreasing unplanned canopy openings (20% of forest area), and explaining that these are areas that were affected by strong wind disturbance in the late 20th century.

5. Implications on Forest Management: The text suggests a shift in forest management towards embracing disturbances rather than solely preventing them, recognizing their natural role in ecosystems (line 169-171). It also stresses the importance of including ecological agents in planning (line 173-174), implying a more proactive approach. However, these perspectives seem inconsistent. To reconcile them, the text could propose integrating reactive and proactive elements, like adaptive management, to respond to disturbances while enhancing resilience for the future.

Response: Thank you for the comment. We have now revisited the text in the light of the comments of the Reviewer, and after careful consideration feel that our text is in fact consistent. The misunderstanding might arise from the term “incorporating disturbance in forest planning”, which the Reviewers seem to have interpreted as proactive approach. In fact what we mean with this is explained in the following sentence, i.e., reducing planned canopy openings and making use of the ones that happen unplanned. We have thus retained the text as we feel that our line of argumentation is in fact consistent.

6. One issue was not satisfactorily resolved: regeneration (comment #5). We pointed out that avoiding discussion of regeneration could be misleading because it conflates sustainable forest management with long-term loss of forest. The authors responded that they addressed regeneration/regrowth/ recovery in a previous paper while the present one focuses on disturbance. In our view, that response does not address the fact that some disturbances have short-term effects with minimal impact (e.g., harvest of a plantation followed by replanting) while others have long-term, high-impact effects (e.g., deforestation resulting from severe disturbance such as fire followed by regeneration failure). This could be dealt with in the manuscript by (1) acknowledging that different post-disturbance trajectories have different effects, and (2) using a snapshot of the data from the previous study, give the reader a general idea of the degree to which regeneration/regrowth/ recovery is occurring. For example, “overall, X% of canopy openings in Europe return to forest cover within 15 years” or whatever the case might be. Additionally, for forest management implications, highlighting or acknowledging management practices aimed at promoting regeneration, such as afforestation, reforestation and silvicultural techniques, is essential. This will help demonstrate how sustainable forest management strategies can mitigate long-term loss for forest cover resulting from severe disturbances (see related comment above on implications to management too).

Response: We've now added the average recovery time (30 years) to the text as suggested, to provide context for the reader. With regard to the silvicultural implications we agree only partly. Specifically, we disagree on the notion that afforestation will help to tackle changing canopy openings. Afforestation, in the mid- to long term, will rather create more closed canopy forests which are prone to disturbance. In fact, the increasing forest area in Europe has been identified as one element that contributes to the currently observed increase in disturbances (see Seidl et al. 2011, GCB doi: 10.1111/j.1365-2486.2011.02452.x). But we do agree that reforestation and silviculture are crucial. In this regard, our text reads "From this we conclude that the focus of forest policy and management should increasingly shift to coping with disturbances rather than aiming to prevent them, e.g., by fostering recovery from and resilience to disturbance."